



# Characteristic study of double substorm onsets in response to IMF variations

Ching-Chang Cheng[1], Christopher T. Russell[2], Ian R. Mann[3], Eric Donovan[4], Wolfgang Baumjohann[5]

[1]Department of Electronic Engineering, National Formosa University, Hu-Wei63201, Taiwan

[2]Department of Earth and Space Sciences and Institute of Geophysics and Planetary Physics, University of California, Los Angeles, CA 90095, USA

[3]Department of Physics, University of Alberta, Edmonton, AlbertaT6G 2J1, Canada

[4]Department of Physics and Astronomy, University of Calgary, Calgary, Alberta T2N 1N4, Canada

[5]Space Research Institute, Austrian Academy of Sciences, Graz 8042, Austria

*Correspondence to*: Ching-Chang Cheng (cccheng@nfu.edu.tw)

**Abstract.** A study of the characteristics of double substorm onsets in response to variations of the interplanetary magnetic

field (IMF) is undertaken with magnetotail and ground observations by the Time History of Events and Macroscale Interactions during Substorms (THEMIS) mission on 18 March 2009 and 3 April 2009 ($Kp \sim 0$), and on 16 February 2008 and 24 February 2010 ($Kp \sim$ 2-3). During the time of interest, THEMIS probes at $-8R_E > XGSM > -20R_E$ and $5R_E > YGSM > -5R_E$ observed earth-bound flow bursts accompanied by magnetic dipolarizations varying in two stages. The keograms and all sky images close to their footprints showed two consecutive auroral breakups of which the first appeared at lower latitudes than

the second. The ground-based magnetometers sensed magnetic bays and perturbations resulting from the formation of substorm current wedge. Two consecutive pulsations in the Pi2-Ps6 band period occurred simultaneously from high to low and very low latitudes. They appeared in the same cycle of growth and then decline in Kyoto-*AL*. The onset timing of ground pulsations mapped to the solar wind observation just in front of Earth's magnetopause shows their occurrence under an IMF variation cycle of north-to-south and then north. Their dynamic spectrums have the spectral features of double substorm

onsets triggered by northward IMF turning. Hence in response to IMF variations, double substorm onsets can be characterized with two-stage magnetic dipolarizations in the magnetotail, two auroral breakups of which the first occurring at lower latitudes than the second, and two consecutive Pi2-Ps6 band pulsations.

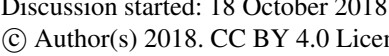



## 1 Introduction

When magnetospheric substorms occur, hydromagnetic disturbances from the Earth's magnetotail to the ground can have bursty bulk flows (or flow bursts), magnetic dipolarizations, the current wedge formation, auroral breakups, high-latitude magnetic bays and Pi2 pulsations in the period range of 40 to 150 s to appear in order (see Baker et al., 1996). Over the past

four decades, two prevailing and competing substorm scenarios, the outside-in (or near-Earth neutral line/point) model (e.g. Hones, 1984) and the inside-out (or current disruption) model (e.g. Lui, 2000), have been proposed as explanations for the development and occurrence sequence of hydromagnetic disturbances in the near-Earth magnetotail during substorms. In addition, Pi2 pulsations can occur successively in a sequence (e.g. Saito, 1969; Clauer and McPherron, 1974) and accompanied by other substorm disturbances in association with the variations of the interplanetary magnetic field (IMF)

(e.g. Cheng et al., 2005). Both models neither agree with the cause of substorm onsets nor explain why substorm disturbances successively occur in a sequence.

On the other hand, with examination of high-latitude magnetograms, Mishin et al. (2001) found that two distinct substorm onsets can have the feature of the first onset occurring at lower latitude than the second one. Mishin et al. (2001) also discovered that they can occur in an IMF variation of north to south and north again. To explain how to lead to double

substorm onsets, Russell (2000) extended the near-Earth neutral point model with emphasis on the role of the distant neutral point. When the incident IMF turns southward, the first onset occurs as the reconnection at the near Earth region starts at the closed magnetic fields and then the second onset does as the reconnection touches the open magnetic fields owing to cessation of the reconnection at the distant Earth region by northward IMF turning. Hence, Russell (2000) suggested that double substorm onsets can occur in a time sequence while the northward IMF turns southward and then northward again.

Moreover, the first onset occurs at lower latitude than the second one as Mishin et al. (2001) reported.

Recently, the THEMIS mission had deployed five Earth-orbiting space probes, all sky imagers and ground-based magnetometers (see Angelopoulos, 2008). The scenario of Russell (2000) under the IMF variation of north-to-south and north again becomes plausible with spectral characteristics and wave mode analyses of consecutive Pi2s during two pairs of double substorm onsets observed by the THMIS mission (e.g. Cheng et al., 2011). Using THEMIS observations, Nakamura

et al. (2011) reported a substorm with two onsets that occurred during a gradual northward IMF turning on 16 February 2008. Based on earthbound fast flows and dipolarization fronts followed by magnetic flux pileup between 8 and 18 $R_E$ in the magnetotail, Nakamura et al. (2011) suggested that activations of reconnection earthward of 18 $R_E$ associated with the first onset make favorable conditions for reconnection tailward of 18 $R_E$ reaching the lobe flux for the second onset. Likewise, Connors et al. (2015) presented a refined view involving a pseudobreakup/expansive/poleward boundary intensification

sequence for the 24 February 2010 substorm through THEMIS observations. They found that the involvement of regions both relatively near the Earth and deeper in the magnetotail can occur in the overall substorm process as the IMF in a variation cycle of north to south and north again. More recently, Cheng et al. (2014) reported that during a geomagnetic quiet period Pi2-Ps6 band pulsations, in a period range from 45 s up to 300 s, can be successively excited by the impulsive sources



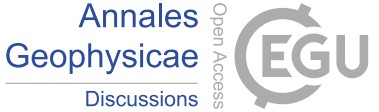

as in Russell (2000) model in response to IMF variations. However, these previous studies did not have all sky images of two successive auroral onsets available for direct verification with other substorm signatures to manifest the physical progress as Russell (2000) suggested. Can they occur first at lower latitudes and then at higher ones? Can they be accompanied by magnetotail dipolarizations varying in two stages? What are exactly the characteristic features for double

substorm onsets?

On 18 March 2009 and 3 April 2009 in the time periods of $Kp\sim$ 0 corresponding to geomagnetic quiet times, and on 16 February 2008 and 24 February 2010, respectively reported by Nakamura et al. (2011) and Connors et al. (2015), in the time periods of $Kp\sim$ 2-3 corresponding to geomagnetic active times, ground magnetic measurements and auroral observations, magnetic and plasma detection by THEMIS probes in the near-Earth magnetotail and other satellites in front of Earth's

magnetopause provide us an opportunity to attack aforementioned questions and undertake the characteristic study of double substorm onsets in response to IMF variations. The $kp$ index is based on the geomagnetic data archived by the world data center for geomagnetism at Kyoto, Japan.

## 2 Data presentation

Note that both 18 March 2009 and 3 April 2009 events are also presented in the manuscript entitled with 'THEMIS

observations of double substorm onsets responsive to IMF variations at geomagnetic quiet times' submitted to *Adv. Space Res.*, and hereinafter referred to as Cheng et al. (submitted, 2018a). In this study, we re-examined flow bursts in the magnetotail, keograms, magnetograms and geomagnetic pulsations on 16 February 2008 and 24 February 2010 events. And the same observational data for quiet-time events are also included to compare to active-time ones.

### 2.1 Locations of THEMIS probes and ground stations

The locations of THEMIS probes and ground stations map for the quiet-time events can refer to Cheng et al. (submitted, 2018a) but for active-time ones are shown in Fig. 1. Figure 1a displays the locations of three selected THEMIS probes in the coordinates of GSM *X-Y* and *X-Z* during 0210- 0310 UT on 16 February 2008. But their locations from 0400 to 0500 UT on 24 February 2010 are displayed in Fig. 1b. Moreover, in Fig. 1a-b, three low-latitude THEMIS magnetometer stations and three very-low-latitude stations, set up by United States Geological Survey (USGS), are marked for reference. On 16

February 2008 and 24 February 2010, THEMIS-A, D and E probes orbited at $XGSM> -10R_E$ (Earth's radii) and $5R_E > YGSM>-5R_E$ in the near-Earth magnetotail. THEMIS-C was at $XGSM> -20 R_E$ and THEMIS-B quite distant from the Earth on 16 February 2008. On 24 February 2010, however, THEMIS-B and C probes were distant from the Earth. Figure 1c-d show the locations of selected THEMIS/CARISMA/Norway and USGS stations marked with red asterisks around the footprints of THEMIS-A with magenta delta-head trace, THEMIS-D with cyan square-head trace, THEMIS-E with blue

cross-head trace obtained with the Tsyganenko (1989) model on 18 March 2009 and 3 April 2009, respectively. Additionally, the footprints of THEMIS-C with green asterisk-head trace are shown in Fig. 1c for the 16 February 2008 event only. The



setup of THEMIS magnetometer stations can refer to Russell et al. (2008). But the magnetometer at LETH is run by the Athabasca University THEMIS UCLA magnetometer network (AUTUMN; http://autumn.athabascau.ca). And the magnetometer at FSMI is set up by the Canadian array for real-time investigations of magnetic activity (CARISMA) (Mann et al., 2008).

**2.2 Flow bursts, keograms, magnetograms and geomagnetic pulsations**

Magnetic and flow burst observations by THEMIS probes, keograms, magnetograms and geomagnetic pulsations around their footprints for quiet-time and active-time events are respectively shown in Figs. 2-3 and 4-5. Figure 2a and b show three components of the ambient magnetic field $B\_GSM$ and the ion flow velocity $Vi\_GSM$ in the magnetotail observed by three selected THEMIS probes from distant to near Earth from 0540 to 0840 UT on 18 March 2009. In this study, sudden

enhancements in ion flow velocity are called flow bursts. We simply map the onset time of ground Pi2s to flow bursts in the magnetotail. Flow bursts are required to meet the criteria set by Angelopoulos et al. (1994) to be segments of continuous ion flow magnitude above 100 km/s and to begin with the velocity exceeding 100 km/s and end as the velocity below 100 km/s. Their onsets are marked with the vertical dashed lines hereinafter. The ambient magnetic field and flow bursts in the magnetotail are in GSM coordinates. +$x$ points to the Earth, +$y$ to the dusk and +$z$ to the geomagnetic north. $Vperp\_x$

denotes the $x$ component of the velocity of flow bursts moving across the ambient magnetic field. Positive and negative $Vperp\_x$ direct earth-bound and anti-earth-bound, respectively. To verify if flow bursts are earth-bound and not field-aligned, the Angelopoulos et al. (1994) approach is adopted to obtain the earthbound component of $Vi\_GSM$ perpendicular to $B\_GSM$ (named $Vperp\_xGSM$ hereinafter). From Fig. 2a and b, one can find that five episodes of field fluctuations like the one affected by magnetic dipolarization occur after the onset of earth-bound flow bursts. By contrast, the fourth field fluctuations

marked with a letter D is the smallest and only seen by THEMIS-D with slight amplitudes. With close inspection of Fig. 2a, one can see that after the last onset of flow bursts, dipolarization-related field fluctuations became enhanced in two stages, clearly seen in the $Bx$ and $Bz$ components at THEMIS-A and marked with E1 and E2. By comparing to the $AL$ index and IMF observations shown in the later section, one can find that a pair of magnetic fluctuations accompanied by dipolarization consecutively occurred in a distinct period of growth and then decline in $AL$ as the IMF turned to south from north and then

north again, corresponding to double onsets in a substorm as Russell (2000) suggested. In this study, multiple hydromagnetic disturbances occurred in a sequence are marked with letters for identification in association with earth-bound flow bursts in the near-Earth magnetotail. Two consecutive onsets of hydromagnetic disturbances related to earth-bound flow bursts during a substorm associated with northward IMF turning are also marked with two serial numbers 1 and 2 for notation. We compared magnetotail observations to ground magnetic and auroral measurements in the same plot format adopted by Lyons

et al. (2013). The top two panels in Fig. 2c display the keograms at SNAP (corrected geomagnetic (CGM) latitude 70.9$^\circ$, $L$=9.34) and FSMI (CGM latitude 67.3$^\circ$, $L$=6.73) observatories from 0754 to 0830 UT on 18 March 2009. There is a clear feature of two consecutive auroral activations, denoted with E1 and E2, related to a double-onset substorm recognized from the $AL$ variation in the later section. As for these two auroral activations, the intensity of the first activation appears to be





stronger than the second one. Moreover, the first auroral breakup occurs at lower latitudes than the second one. The middle three panels in Fig. 2c show three components of the magnetic field at SNAP, FSMI and LETH (CGM latitude 56.9°, $L$=3.35) near the footprints of THEMIS probes. Two magnetic bays in $H$ at SNAP occurred after E1 and E2 onsets. The magnitude of $Z$ at SNAP seemed not changed for three minutes after E1 and then sharply decreased before the E2 onset. After onset E2,

the magnitude of $Z$ at SNAP turned to increase. These observational results are consistent with the Mishin et al. (2001) study. In addition, the $H$ component at LETH enhanced after E1 and persisted through E2. The last two panels in Fig. 2c show three components of magnetic pulsations at SNAP and LETH that are 40 s running averaged. One can see that magnetic pulsations at SNAP are in the period range of Pi2-Ps6 and those at LETH in Pi2 instead.

In the same format as Fig. 2, Figs. 3, 4 and 5 show the similar observational results for 3 April 2009, 16 February 2008,

and 24 February2010 events, respectively. Figure 3a and b show three components of $B\_GSM$ and $Vi\_GSM$ measured by three selected probes from the distant Earth region to near Earth one from 0645 to 0745 UT on 3 April 2009. One can see from Fig. 3a and b that resembling the 18 March 2009 event, the 3 April 2009 event has three successive field fluctuations like the one affected by magnetic dipolarization after activations of earthbound flow bursts. Moreover, in Fig. 3a, the last field fluctuations seem to perturb with two consecutive pulses, in which one has a shorter period and the other a longer one,

clearly seen in $Bx$ and $Bz$ at THEMIS-D and marked with C1 and C2 for reference. The keograms at SNAP and FSMI from 0645 to 0745 UT on 3 April 2009 are displayed in Fig. 3c with the plot format like Fig. 2c. One can find from the top two panels in Fig. 3c that two distinct auroral activations marked with C1 and C2 appear to occur in a sequence. The middle three panels in Fig. 3c show three components of the magnetic field at SNAP, FSMI and near the footprints of THEMIS probes. Two magnetic bays in $H$ at SNAP occurred after C1 and C2 onsets. The magnitude of $Z$ at SNAP had a slight increase after

C1 and later sharply decreased before the C2 onset. After onset C2, the magnitude of $Z$ at SNAP increased. The last two panels in Fig. 3c show three components of magnetic pulsations in Pi2-Ps6 at SNAP and in Pi2 at LETH. These observational features in the 3 April 2009 event look the same as the 18 March 2009 event. As a result, accompanied by earth-bound flow bursts in the near-Earth magnetotail, two-stage magnetic dipolarizations in the magnetotail, two auroral breakups of which the first occurring at lower latitudes than the second, and two consecutive Pi2-Ps6 band pulsations can

occur at geomagnetic quiet times.

In the same format as Fig. 2, Fig. 4 shows the similar observational results at active times with $Kp$~ 3 for the 16 February 2008 event reported by Nakamura et al. (2011). One can find from Fig. 4a and b that flow bursts occur recurrently with three magnetic dipolarizations of which the last one appears to vary in two stages, marked with C1 and C2, clearly seen at THEMIS-D and E. The keograms at RANK and KUUJ from 0210 to 0310 UT on 16 February 2008 are displayed in Fig. 4c

with the plot format like Fig. 2c. One can find from the top two panels in Fig. 4c that in addition to two auroral activations only at KUUJ, two others marked with C1 and C2 appear to first occur at KUUJ and then at RANK in a sequence. The middle panels in Fig. 4c shows that magnetic bay in $H$ at RANK occurred after 0242 UT when the $D$ component at CHBG suddenly increased. The last two panels in Fig. 4c show three components of magnetic pulsations in Pi2-Ps6 at RANK and in



Pi2 at CHBG. These observational results suggest that there were two pseudosubstorm onsets at 0218 and 0230 UT followed by two substorm onsets at 0242 and 0251 UT.

In the same format as Fig. 2, Fig. 5 shows the similar observational results at active times with $Kp$~ 3 for the 24 February 2010 event reported by Connors et al. (2015). One can find from Fig. 5a and b that magnetic dipolarization appears to have a two-stage variation accompanied by flow bursts, marked with #1 and #2, clearly seen at THEMIS-D and E. The keograms at RANK and GILL from 0400 to 0500 UT on 24 February 2010 are displayed in Fig. 5c with the plot format like Fig. 2c. One can find from the top two panels in Fig. 5c that auroral activations feature a pseudobreakup followed by two breakups first occurring at GILL at 0426 UT and then at RANK at 0438 UT in a sequence. The middle panel in Fig. 4c shows that magnetic bay in $H$ at RANK occurred after 0426 UT when the $D$ component at PINE suddenly increased. The last two panels in Fig. 5c show three components of magnetic pulsations in Pi2-Ps6 at RANK and in Pi2 at PINE. These observational results suggest that there were a pseudobreakup at 0411 UT followed by two substorm onsets at 0426 and 0438 UT.

### 2.3 All sky images

For further verification of auroral breakups found from the keograms, we also checked out THEMIS all sky images in the movie and mosaic archives during the time of interest. All sky images of aurora around footprints of THEMIS probes for the quiet-time events can refer to Cheng et al. (submitted) but for active time events are respectively shown in Figs. 6-7. From the movie archives (accessible at http://themis.ssl.berkeley.edu) not shown in this study, one can found that auroral breakup appeared and gradually intensified over KUUJ observatory after 02:42:33 UT corresponding to the C1 onset in contrast to pre-existed auroral brightenings seen by SNKQ observatory on 16 February 2008. This observational feature can be seen from the mosaics at the pre-onset time 02:42:12 UT and the post-onset time 02:42:36 UT shown in Fig. 6a and b, respectively. Before 02:50:51 UT, auroral brightenings existed at the lower part of sky image over RANK and suddenly intensified and accompanied by westward and eastward expansions after 02:51:06 UT corresponding to the C2 onset on 16 February 2008. Figures 6c and d show that the mosaics at the pre-onset time 02:50:51 UT and the post-onset time 02:51:06 UT have the same feature found from the movie archives. As for the 24 February 2010 event, Fig. 7a and b show that auroral activations were not obvious before 04:11:00 UT and then gradually intensified but not strong until 04:12:00 UT corresponding to pseudobreakup (PS). Figure 7c and d show that before 04:27:30 UT, auroral brightenings existed over GILL and suddenly intensified and accompanied by poleward expansions after 04:27:33 UT corresponding to the #1 onset on 24 February 2010. Figure 7e and f show that before 04:38:03 UT, auroral activations moved poleward from the upper part of sky image at GILL to the lower part of sky image at RANK and then suddenly intensified and accompanied by westward and eastward expansions after 04:38:06 UT corresponding to the #2 onset. As a result, as in the quiet-time events studied by Cheng et al. (submitted), THEMIS all sky images show the same observational feature found from keograms that auroral breakups appear to occur at lower latitudes for the onset 1 than the onset 2 for the active-time events.



### 2.4 Ground magnetic measurements at mid and very-low latitudes

Ground magnetic measurements at very low latitudes for quiet time events can refer to Cheng et al. (submitted) but for active time events are respectively shown in Fig. 8. We check out the magnetograms at mid and very-low latitudes to see if their magnetic variations were affected by the formation of substorm current wedge (SCW). Figure 8a show the $H$ and $D$ components at THEMIS/Norway mid-latitude stations longitudinally spread to the east from the west during 0210-0310 UT on 16 February 2008. Figure 8a also displays that the $H$ component at all five stations had wave-like perturbations at the A and B onsets but at KAPU and CHBG began to increase with small amplitude perturbations at the C1 onset and with larger amplitude after the C2 onset. And these periodic perturbations remained until 0300 UT except for DOB (CGM longitude 59.3°, latitude 89.6°, $L$=3.85) and DON (CGM longitude 63.4°, latitude 94.6°, $L$=5.00). Figure 8a shows that the $D$ component has two distinct bay variations in which the first one is positive at TPAS, KAPU and CHBG and the second negative at DOB and DON. Except for DOB and DON, magnetic pulsations can be clearly seen in $D$ at most stations. The magnetic bay variation began at the C1 onset and became stronger after the C2 onset. Figure 8b show the $H$ and $D$ components at the very-low-latitude stations set up by the United States Geological Survey (USGS) longitudinally spread to the east from the west during 0400-0500 UT on 24 February 2010. One can find from Figure 8b that the $D$ component at FRN, TUC and BSL had a positive bay variation and that at BSL (CGM longitude 341.5°, latitude 41.0°, $L$=1.76) a negative one after the #1 onset. Referring to Clauer and McPherron (1974), lower-latitude magnetometers below the upward field-aligned current can record a positive bay variation and those below the downward field-aligned current a negative one. Hence, magnetic variations detected by selected mid- and very-low-latitude stations can be the ones influenced by the SCW formation for these selected events. In other words, the SCW formation can occur at double onsets in both 16 February 2008 and 24 February 2010 events.

### 2.5 Solar wind observations and geomagnetic indices

Solar wind observations and geomagnetic indices for quiet time events can refer to Cheng et al. (submitted, 2018a) but for active time events are respectively shown in Fig. 9. We browse the OMNI combined data to see if successive substorm-related disturbances respond to the incident IMF variations. The incident IMF in the upstream region, sensed by the ACE satellite at XGSM ~240 $R_E$, is imitated as just outside the Earth's bow shock atXGSM~17.0$R_E$ using minimum variance analysis (e.g. Weimer et al., 2003). As shown in the top five panels of Fig. 9a, the IMF-$By$, $Bz$, clock angle, the solar wind speed ($Vsw$) and the solar wind dynamic pressure ($Dp$) measured by ACE in the upstream region are time shifted to the nose of the Earth's bow shock (named as OMNI data hereinafter) and by Geotail, orbited at the Earth's magnetosheath in the dawn sector, at XGSM ~ -4.0 $R_E$ from 0100 to 0400 UT on 16 February 2008. Mapping the onset time of substorm-related disturbances to the OMNI data, one can find from Fig. 9a that the time-shifted IMF appears to have the same variation as seen by Geotail with larger magnitudes. The IMF $Bz$ at Geotail varied with a clearer orientation cycle that turns to south from north and then back to north than OMNI having slight magnitude variation along the zero line. Moreover, both OMNI





and Geotail sensed positive IMF *By* for each onset. Owing to Geotail having stronger IMF *By* than OMNI, the IMF clock angle remained 90 degree for OMNI and -90 degree for Geotail before the C2 onset. But as the IMF turned north after the C2 onset from south at the C1 onset, both had decreasing magnitude in the IMF clock angle. This shows the orientation change of the IMF to south from north and then north again that is the same as those found by Cheng et al. (2011). Moreover, *Vsw*

and *Dp* stayed around 650 km/s and 1.8 nPa, respectively. Their magnitudes do not have any sharp change then. In Fig. 9a, the bottom two panels display that Kyoto-*AL* was above -300 nT and *SYM-H* higher than -15 nT. Both C1 and C2 onsets can be double onsets responsive to the IMF varying to south from north and back to north as Russell (2000) suggested.

Figure 9b, with the plot format like Fig. 9a, shows the incident IMF *By* and *Bz* components by OMNI and Cluster 1 at XGSM ~ 15.0 $R_E$, OMNI-*Vsw* and OMNI-*Dp*, and geomagnetic indices from 0300 to 0500 UT on 24 February 2010. For

OMNI and Cluster 1, the IMF *By* remained positive during the onsets of substorm-related disturbances. As for the IMF *Bz*, both OMNI and Cluster 1 have a clearer variation of north-to-south and north again prior to the #2 onset. As for the IMF clock angle, both had a similar variation, first north then to south and north again, albeit with positive magnitude. OMNI-*Vsw* and OMNI-*Dp* stayed around 360 km/s and 1.5 nPa with little sharp change. Figure 9b shows that Kyoto-*AL* remained above - 300 nT and *SYM-H* over -10 nT. Hence for the 24 February 2010 event, the occurrence of two consecutive substorm-

related disturbances can result from magnetotail reconnection with reflection to the IMF variations as Russell (2000) suggested.

As a result, as Cheng et al. (submitted, 2018a) found from the quiet-time events, two consecutive substorm-related disturbances can occur in association with an IMF variation to south from north and back to north again under geomagnetic active intervals while *Vsw* and *Dp* remain unchanged.

**2.6 Dynamic spectrums of ground pulsations**

Dynamic spectrums of ground pulsations for quiet time events can refer to Cheng et al. (submitted, 2018a) but for active time events are shown in Fig. 10. In order to have the temporal variation of pulsation frequency, we follow Cheng et al. (submitted, 2018a) to perform the wavelet transform of *dH/dt* and *dD/dt*. Figure 10a shows the dynamic spectrums of *dH/dt* and *dD/dt* at THEMIS/CARISMA ground stations along the meridian near the footprints of three selected THEMIS probes

near East Coast from 0210 to 0310 UT on 16 February 2008. Three significant frequencies at 2, 4 and 6 mHz are denoted with horizontal dashed lines. Figure 10a also shows that for A and B onsets, the significant spectral magnitude in *dH/dt* was around 6 mHz from RANK down to CHBG except GBAY and CHBG up to 10 mHz. For C1 onset, two harmonic significant frequencies are around 10 mHz and between 4-6 mHz (i.e. Pi2 band) at all stations. But after C2 onset, the significant spectral frequencies tend to decrease to ~ 2-6 mHz (i.e. Pi2-Ps6 band) from around 10 mHz. As for *dD/dt*, the dynamic

spectrums are shown in Fig. 10b. From Fig. 10b, one can see that *dD/dt* has the same significant spectral magnitudes as in *dH/dt* for A and B onsets. For C1 to C2 onsets, *dD/dt* has the same trend of decreasing dominant frequency as in *dH/dt*. In the same format as Fig. 10a, Figure 10c and d show the dynamic spectrums of *dH/dt* and *dD/dt* at THEMIS ground stations along Churchill line near the footprints of three selected THEMIS probes from 0400 to 0500 UT on 24 February 2010.



Likewise, one can see from Figure 10c-d that both *dH/dt* and *dD/dt* has the same trend of decreasing dominant frequency as in *dH/dt* from #1 to #2 onsets. As Cheng et al. (submitted, 2018a) found from the quiet-time events, these spectral analyses suggest that for two consecutive pulses at active times from auroral to lower latitudes, the dominant frequency of onset 1 seems to be Pi2-band higher than that of onset 2 being Ps6-band.

**3 Discussion and summary**

In this study, we have presented well-coordinated observations of hydromagnetic disturbances including remarkable auroral activations in double substorm onsets by the THEMIS mission at geomagnetic quiet and active times. During the interested times, THEMIS probes, at $-8R_E > XGSM > -20R_E$ and $5R_E > YGSM > -5R_E$, observed recurrent magnetic dipolarizations (see Figs. 2-5) in which the last one varied in two stages that is significant for justification of double onsets as Russell (2000) proposed.

And earth-bound flow bursts (see Fig. 2) are similar to the findings by (Baumjohann et al., 1990) that flow bursts are mostly earth-bound and responsible for momentum and flux transport expectedly excited by magnetic reconnection in the magnetotail.

The keograms at RANK (*L*=10.84) and KUUJ (*L*=6.50) near their footprints showed successive auroral activations (see Figs. 4-5), further verified with the mosaics of THEMIS all sky images (see Figs. 6-7), that are consistent with earlier reports

by Fairfield et al. (1999) and Nakamura et al. (2001) that auroral breakups subsequently occurred near their footprints while earthbound flow bursts were detected in the magnetotail. Noteworthy, auroral breakups for onset 2 occur at higher latitude than for onset 1. In the meantime, the magnetometers at mid and very low latitudes sensed magnetic deflections like the ones influenced by the SCW formation (see Fig. 8) that is the same as Clauer and McPherron (1974) suggested. Moreover, two consecutive pulsations in the Pi2-Ps6 band period occurred simultaneously from high latitudes down to very low latitudes

(see Figs. 2-5). These observational results support the suggestion by Shiokawa et al. (1998) that braking of earth-bound flow bursts leads to Pi2 onset and the earlier finding that Ps6 can occur right after the Pi2 onset in a time sequence (Cheng et al., 2014 and references of therein).

Since two consecutive Pi2-Ps6 band pulsations and magnetic bays are coincident with auroral breakups, they can result from the flow bursts in the magnetotail as Kepko et al. (2004) suggested. During substorm-related activations, reconnection-

driven plasma injections or flow bursts in the magnetotail can excite hydromagnetic disturbances to propagate earth-bound in fast waves and couple to local field lines in Alfvén waves (e.g. Cheng et al., 1998, 2004, 2009). Hence, their source site and travelling route can determine the dominant frequency of wave-driven geomagnetic pulsations which relies on the cavity extent, the field line span and the ambient plasma density. According to Russell (2000), onset 1 can occur as the reconnection in the near Earth magnetotail initiates in the closed magnetic fields preceded by southward IMF turning and

subsequently onset 2 can appear as the reconnection further reaches the open magnetic fields after discontinuation of the reconnection in the distant Earth magnetotail by northward IMF turning. This suggestion is supported by the observations of magnetic dipolarization varying in two stages (see Fig. 2) and all sky images of auroral breakups for onset 2 occurring at





higher latitude than onset 1 (see Fig. 4). The onsets of substorm-related activations mapped to the incident solar wind at the nose of the Earth's magnetopause demonstrate their occurrence in a time sequence while the northward IMF turns southward and then northward again (see Fig. 9). Using wavelet transformation of *dH/dt* and *dD/dt* for the active-time events as in Cheng et al. (submitted, 2018a), their dynamic spectrums have the spectral features of double substorm onsets triggered by

northward IMF turning (see Fig.10).

Like Cheng et al. (submitted, 2018a), the present study shows new observational findings for the last two onsets on 16 February 2008 having a two-stage variation of magnetic dipolarization in the magnetotail, two auroral breakups of which the first occurring at lower latitudes than the second, and two consecutive Pi2-Ps6 band pulsations. Likewise, we have done the similar analysis of the 24 February 2010 event during the geomagnetic active time with *Kp* ~ 2 studied by Connors et al.

(2015). The comparison of geomagnetic active and quite events of two substorm onsets in response to the IMF variations shows that the occurrence sequence of all required substorm signatures looks the same and not different for small and large *Kp.*.

We note that previous statistical studies (Morley and Freeman, 2007; Freeman and Morley, 2009; Newell and Liou, 2011) test if every northward IMF turning leads to substorm onset. With analysis of their selected data, they conclude that a period

of southward IMF and magnetospheric loading is a necessary condition for substorm to occur but not for northward IMF. Hence, they argue that substorms appear to be a magnetospheric internal process and northward IMF turning is not necessary for substorm triggering. However, their conclusions come from single-satellite observations with an assumed propagation time delay and based on satellite global images for identification of substorm onset only. With different criteria of event selection from previous works, this study, following Cheng et al. (submitted, 2018a), adopts more other substorm signatures

from THEMIS space and ground-based measurements for identification and focuses on two consecutive onsets occurring in a cycle of growth-to-decay in the *AL* index and under an IMF variation cycle of north-to-south and north again. The first substorm onset does not have to occur after northward IMF turning. For justification, as in Cheng et al. (submitted, 2018a), we have performed correlation analysis of IMF *By* and *Bz* in the upstream region versus those at the dayside Earth's magnetopause for the active-time events (see Fig. 11).

Hence in response to IMF variations, double substorm onsets can be characterized with two-stage magnetic dipolarizations in the magnetotail, two auroral breakups of which the first occurring at lower latitudes than the second, and two consecutive Pi2-Ps6 band pulsations

*Data availability.* THEMIS, OMNI, and Cluster data used in this paper were obtained from CDAWeb

http://cdaweb.gsfc.nasa.gov/istp_public/. THEMIS all sky images and movie archives were also accessible at the website http://themis.ssl.berkeley.edu.

*Author contributions.* CC designed and analyzed the work, and wrote the paper. CR, IM, ED, and WB set up THEMIS measurements and commented on the paper.



*Competing interests.* The authors declare that they have no conflict of interest.

*Acknowledgments.* The THEMIS data were obtained via CDAWeb and the work at UCLA was supported by NASA under
the grant UCB/NASA 5-02099.The AUTUMN magnetic array is operated by M. Connors at Athabasca University with
installation funded by the Canada Foundation for Innovation. The CARISMA magnetic array is run by the University of
Alberta and funded by the Canadian Space Agency. The USGS magnetic field data were provided by C. Finn at USGS via
CDAWeb. The THEMIS GBO all sky imagers are jointly operated by S. Mende at UC Berkeley. The OMNI combined data
were provided by J.H. King and N. Papatashvilli at NASA GFSC via CDAWeb. The magnetic data at Cluster 1were
provided by A. Baloghat ICSTM via CDAWeb. The Geotail magnetic fields were provided by DART at ISAS/JAXA. This
work was supported by Ministry of Science and Technology of R. O. C. on Taiwan under the grant MOST105-2111-M-150-
001.

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





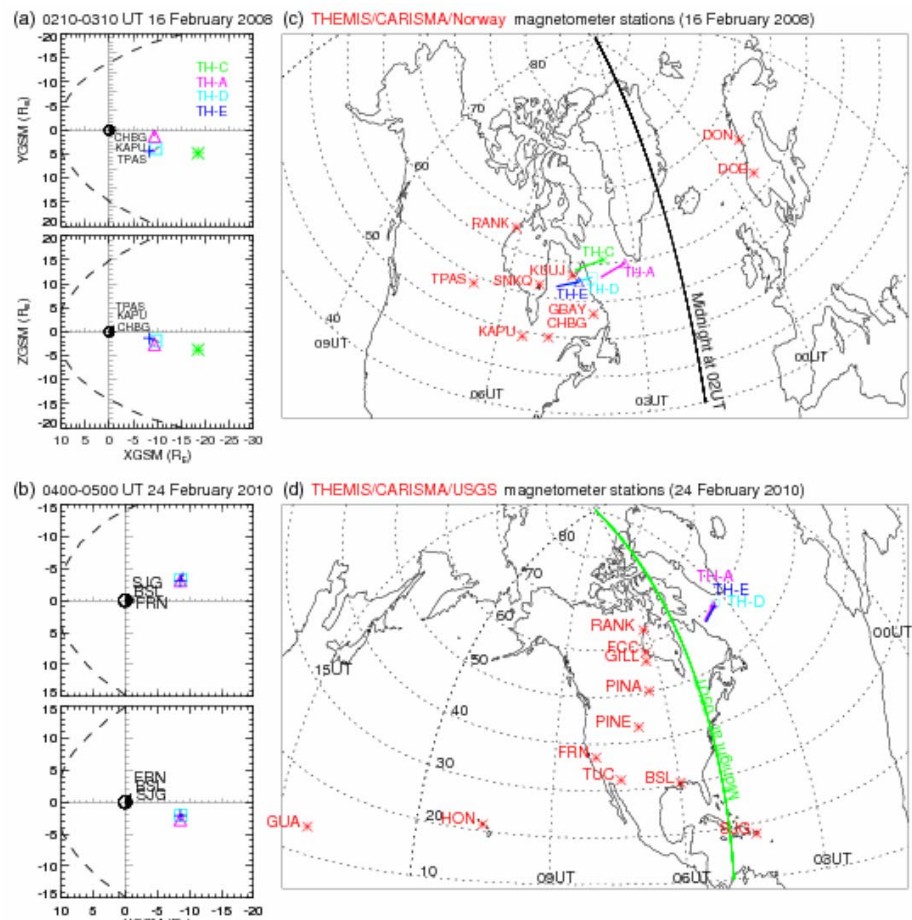

**Figure 1:** (a) The locations of THEMIS-A, C, D and E probes in GSM *X-Y* and *X-Z* from 0210 to 0310 UT on 16 February 2008. Three ground stations at mid-latitudes are marked for reference. (b) In the same format as Fig. 1a, except from 0400 to 0500 UT on 24 February 2010. (c) The locations of the THEMIS/CARISMA/Norway (red asterisks) magnetometer stations, the footprints of THEMIS-C (green trace with square head), THEMIS-D (cyan trace with square head), THEMIS-E (blue trace with cross head) and THEMIS-A (magenta trace with delta head) for reference on 16 February 2008. (d) In the same format as Fig. 1c, except for the THEMIS/CARISMA/USGS magnetometer stations on 24 February 2010.




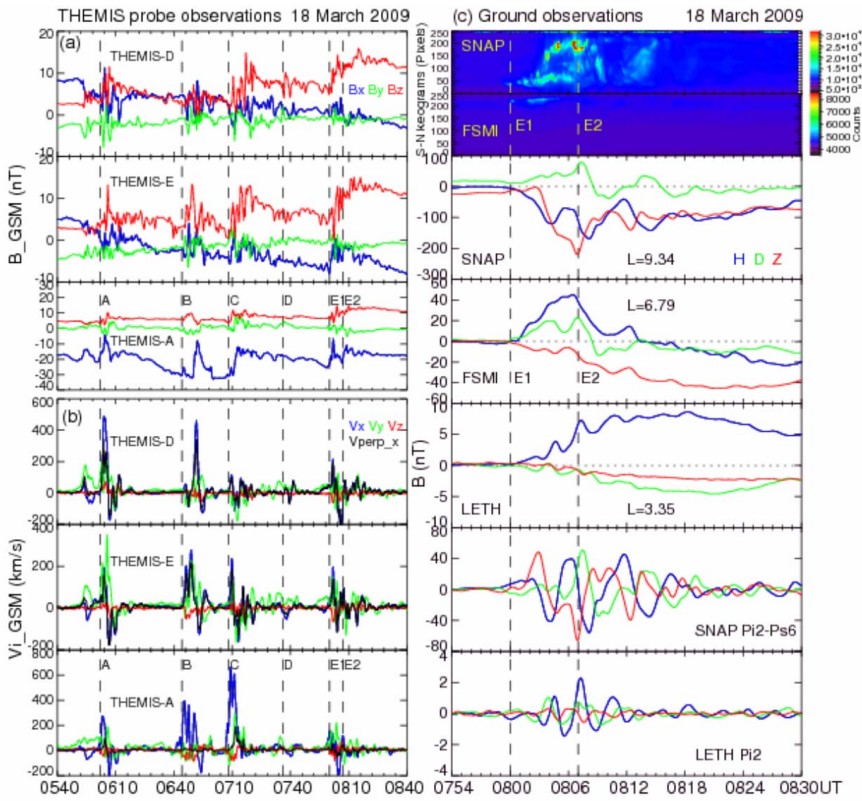

**Figure 2:** (a) Three components of the magnetic fields sensed by the selected THEMIS probes from 0540 to 0840 UT on 18 March 2009. The vertical dashed lines denote the onsets of earth-bound flow bursts marked with a letter for identification. Under a substorm cycle associated with northward IMF turning, two consecutive earth-bound flow bursts are also marked with serial numbers 1 and 2. (b) In the same format as Fig. 2a, except for the ion flow velocity. (c) Keograms, three components of the magnetic fields and pulsations at SNAP, FSMI and LETH from 0754 to 0830 UT on 18 March 2009.





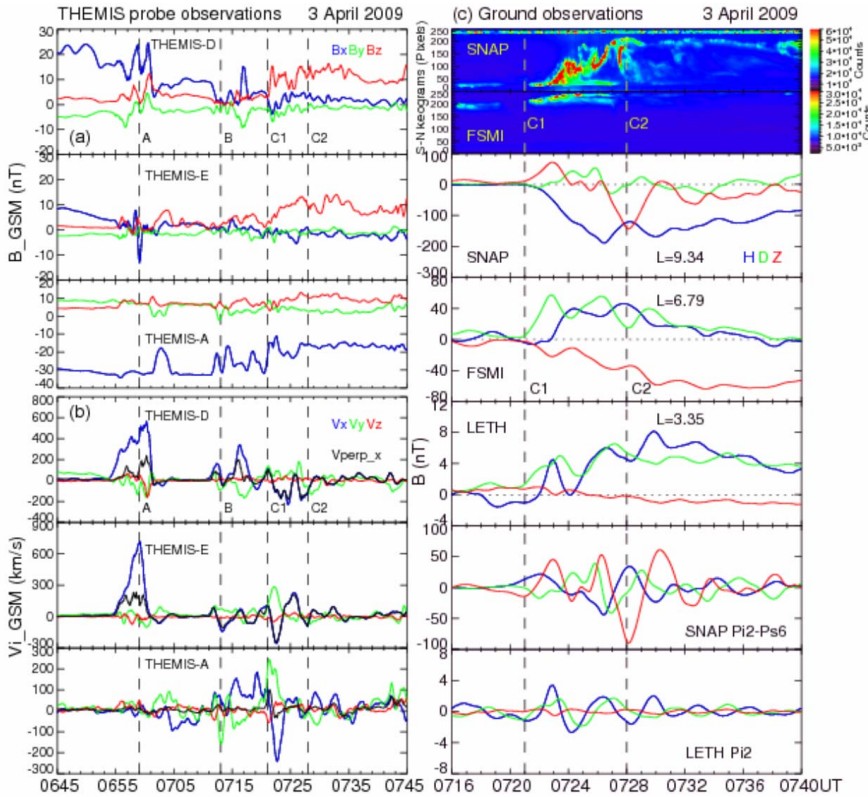

**Figure 3:** (a-b) In the same format as Fig. 2a and b, except from 0645 to 0745 UT on 3 April 2009. (c) In the same format as Fig. 2c, except for SNAP, FSMI and LETH from 0716 to 0740 UT on 3 April 2009.





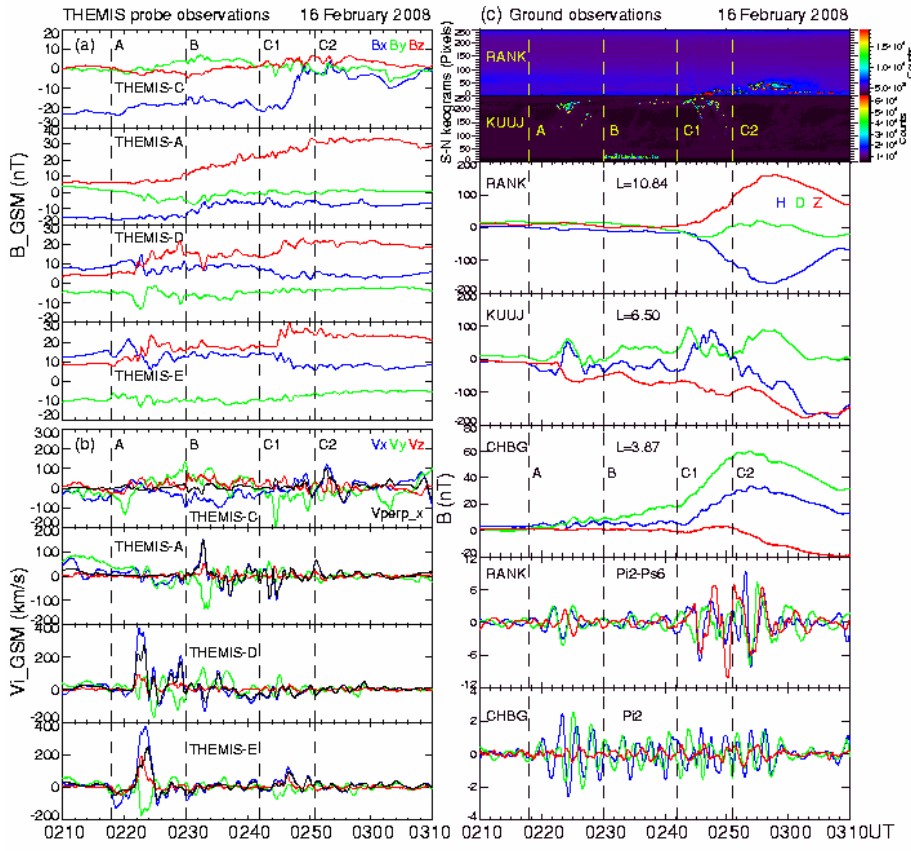

**Figure 4:** (a-b) In the same format as Fig. 2a and b, except from 0210 to 0310 UT on 16 February 2008. (c) In the same format as Fig. 2c, except for RANK, KUUJ and CHBG from 0210 to 0310 UT on 3 April 2009.





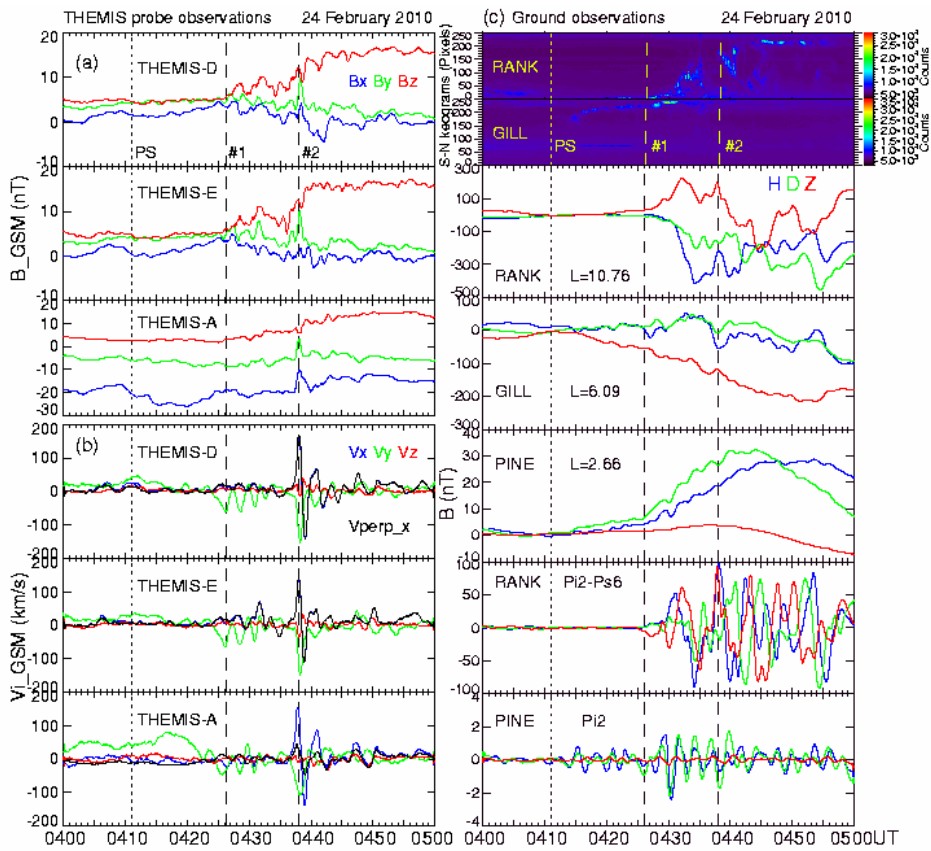

**Figure 5:** (a-b) In the same format as Fig. 2a and b, except from 0400 to 0500 UT on 24 February 2010. The vertical dotted line denotes the onset of auroral pseudobreakup (PS). The vertical dashed lines denote the onsets of earth-bound flow bursts marked with serial numbers 1 and 2. (c) In the same format as Fig. 2c, except for RANK, GILL and PINE from 0400 to 0500 UT on 3 April 2009.





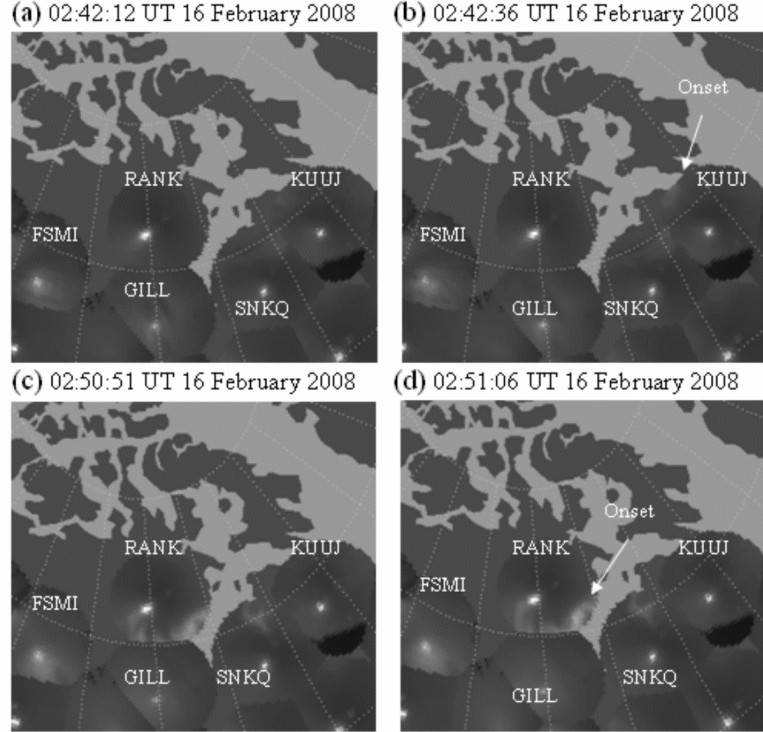

**Figure 6:** (a-b) THEMIS all sky images related to the C1 onset on 16 February 2008. The arrows denote auroral breakup onsets. FSMI, RANK, GILL, SNKQ and KUUJ observatories are marked for reference. (c-d) In the same format as Figure 6a, except for relating to the C2 onset.



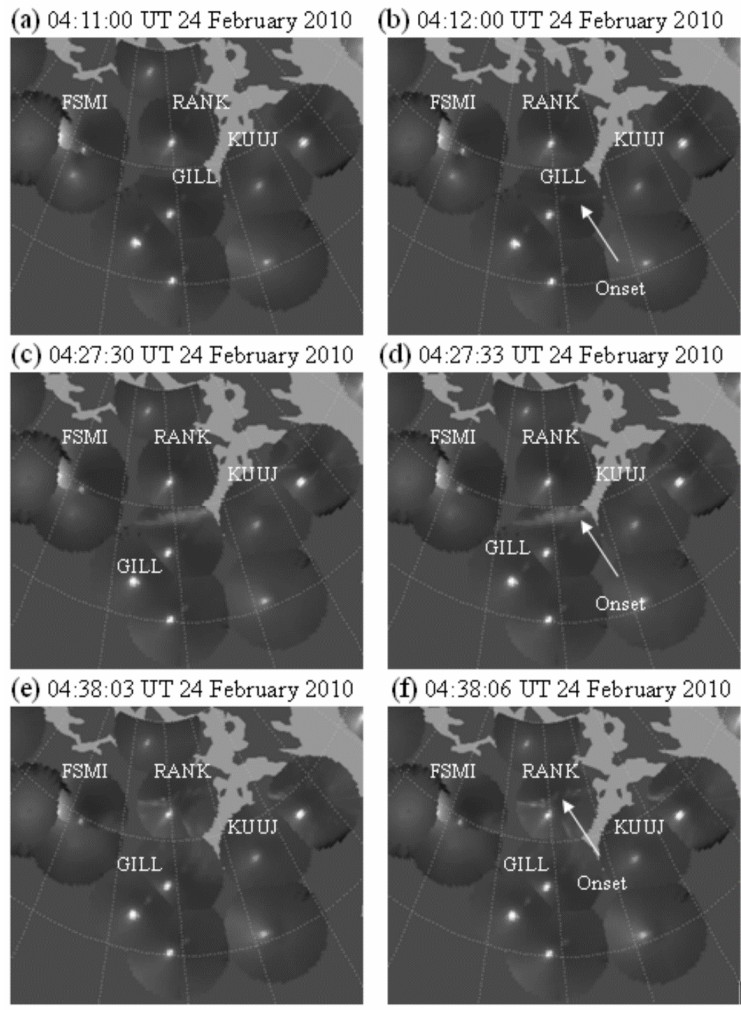

**Figure 7:** In the same format as Figure 6, except for on 24 February 2010.





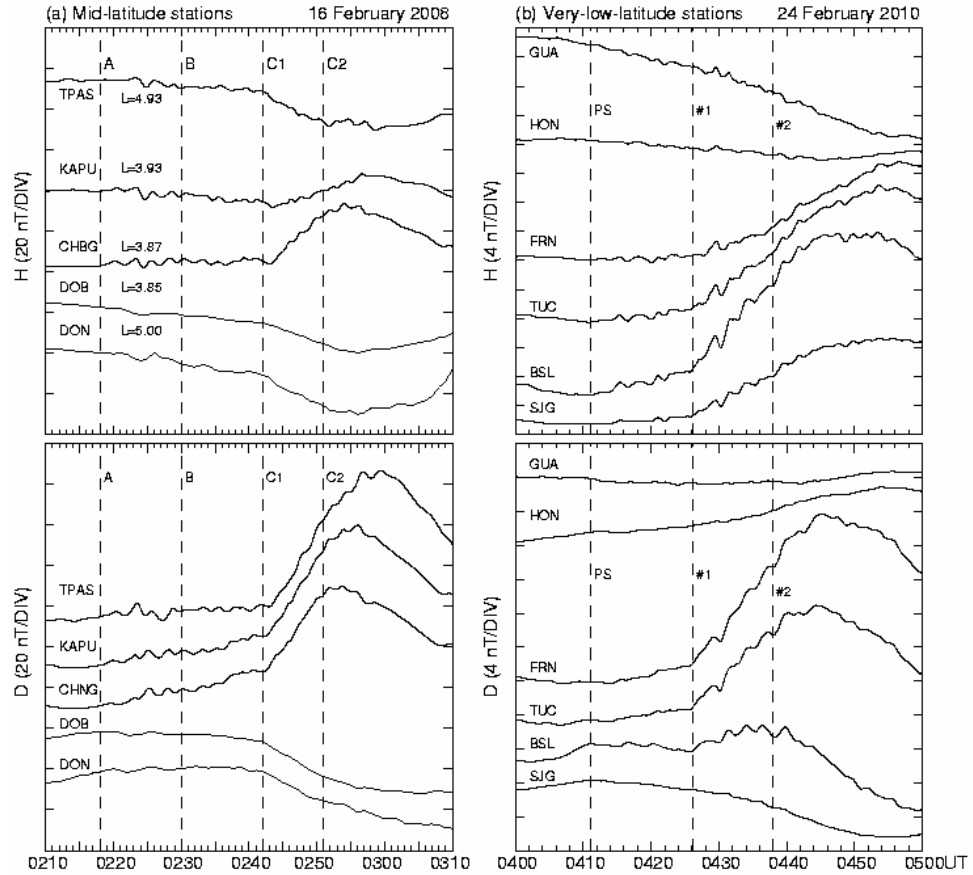

**Figure 8:** (a) The *H* and *D* components at mid-latitude stations across the Atlantic Ocean from 0210 to 0310 UT on 16 February 2008. The vertical dashed lines denote the onsets of earth-bound flow bursts in the plasma sheet. (b) In the same format as Fig. 8a, except from 0400 to 0500 UT on 24 February 2010.





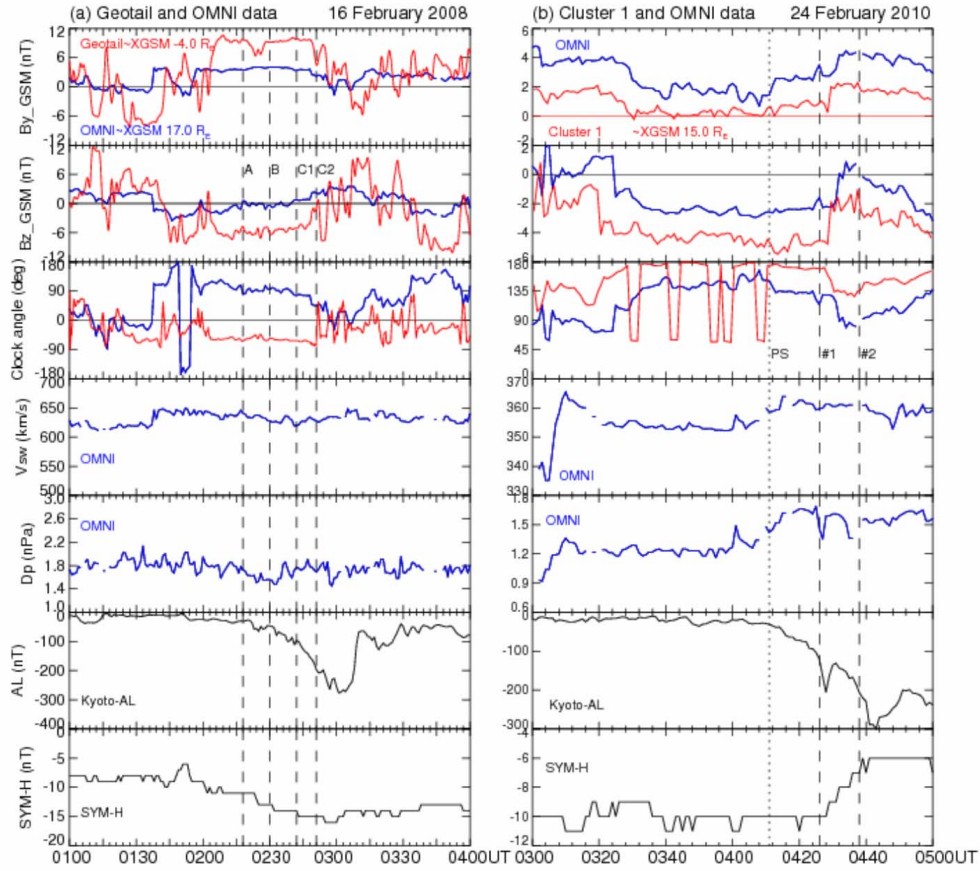

**Figure 9:** (a) The IMF-*By*, *Bz* and clock angle by OMNI (corresponding to ~XGSM 17.0 $R_E$) and Geotail (~XGSM -4.0 $R_E$), the solar wind speed *Vsw* and the solar wind dynamic pressure *Dp* by OMNI, Kyoto-*AL* and *SYM-H* indices from 0100 to 0400 UT on 16 February 2008. (b) In the same format as Fig. 9a, except for Cluster 1 (~XGSM 15.0 $R_E$) and OMNI from 0300 to 0500 UT on 24 February 2010.





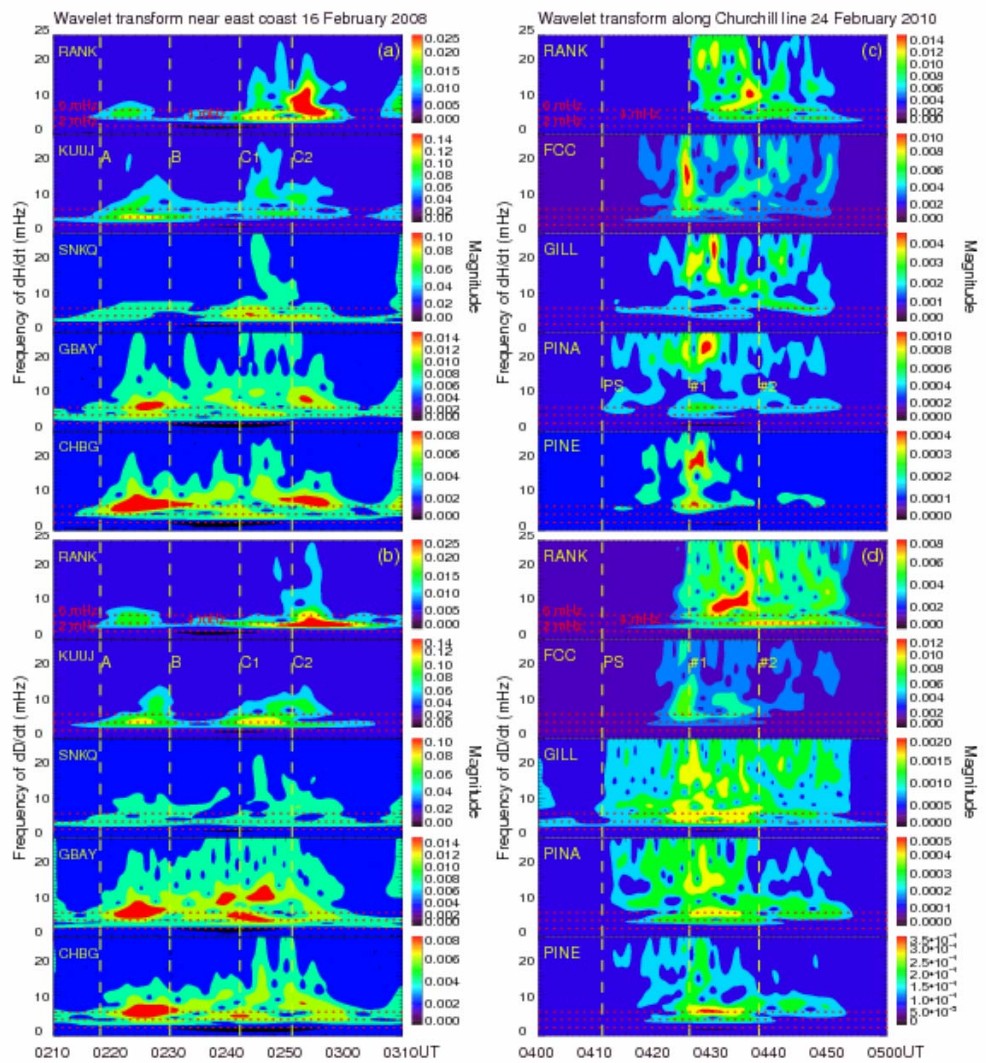

**Figure 10:** (a-b) The wavelet transform of *dH/dt* and *dD/dt* at the THEMIS stations near East Coast for the 16 February 2008 event. (c-d) In the same format as Fig. 10a-b, except at the THEMIS stations along Churchill line for the 24 February 2010 event.





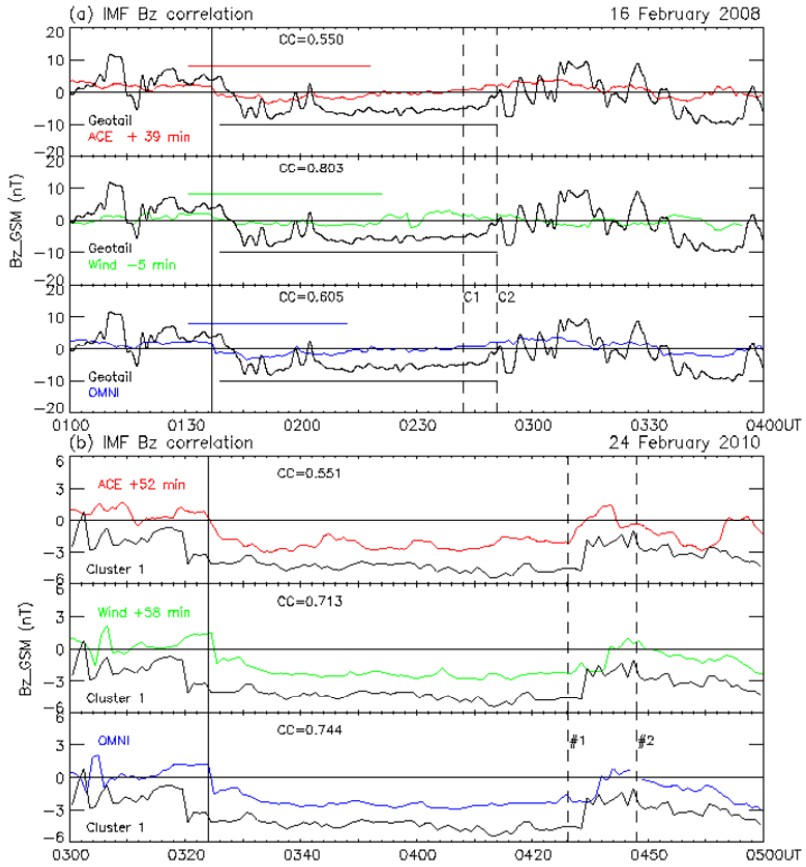

**Figure 11: (a)** The correlation analysis of IMF *Bz_GSM* by ACE, Wind and OMNI versus Geotail from 0100 to 0400 UT on 16 March 2008. The two vertical dashed lines denote double substorm onsets. The vertical line denotes southward turning of the IMF at ACE, Wind and OMNI The horizontal lines mark the time intervals of north to south and north again for correlation analysis. **(b)** In the same format as

5 Fig. 11a, except for Cluster 1 and the time of interest from 0300 to 0500 UT on 24 February 2010.