# Peer review of "Characteristic study of double substorm onsets in response to IMF variations"

_Annales Geophysicae, 2018_

## Referee Comment (RC1) · Anonymous Referee #1 · 30 Nov 2018

This paper describes four substorms that the authors describe as double onset substorms. All four events have been analyzed in previous publications, two of them (which receive cursory treatment here) in a paper recently submitted to another journal by the first author, the other two by different authors. This study adds auroral and ground magnetometer data to what was presented in the previous papers on the other two events. The authors claim that these substorms are examples of double onset substorms, and that the second onset occurs poleward (and hence tailward) of the first onset and that it is triggered by a northward turning of the IMF. I did not find the evidence for this claim, as presented in the figures, very convincing. I also find that there is insufficient that is new in this paper to warrant publication.

Substorms are complex natural events involving magnetic reconnection in what we are

[Figure]

learning is a highly structured geomagnetic tail. Like most complex natural events, no two substorms are the same, nor should we expect them to be, but they do have many features in common. Many if not most substorms include multiple auroral brightenings that can both precede (pseudobreakups) and follow (intensifications) the primary onset of the substorm. These are usually accompanied by Pi2 pulsations observed on the ground and by fast flow bursts and dipolarizations in the tail if a spacecraft happens to be in the right position to observe them. (Part of the problem with this field is that the terminology is not universally agreed upon, but I have tried to use commonly used terms.) This is the case with the four events presented here.

Figures 2-5 present summary plots of the four events. (For two of the events this is in only data presented.) The authors do not explain how the times of the vertical lines marking particular events were chosen, nor is it obvious from the data (which is hard to make out given the size of the figures). However, the only line important to this paper's argument is the last (rightmost) one in each figure, and the important data is presented in the righthand panel. In each case after the initial breakup the aurora expand poleward and include a number of brightenings. The magnetometer traces have clear bays and Pi2s accompanying the initial onset, but thereafter contain many variations confused by the waves also present. I could see no features in either the auroral or magnetometer data that clearly differentiate the time of the second vertical line. Figures 6 and 7 present the auroral data for the second two events, concentrating on the times of auroral brightenings. The presentation of the figures is far from ideal; I found it very difficult to see much in them. However the auroral keograms in figures 4 and 5 show that there are other auroal brightenings not shown here. Figure 9 presents mid and low latitude magnetometer data. These show the clear bays accompanying the initial onset and a number of Pi2s, but again I see no particular features associated with the rightmost line.

The evidence for the second conclusion, that the second onsets are triggered by northward turnings of the IMF, is contained in figures 9 and 11. Here the authors run into the

notorious problems of the lack of uniformity of the IMF (different spacecraft see some-what different signatures) and of timing the IMF's arrival at the magnetopause and its subsequent effects in the tail. All I can say is that I found the evidence presented here to be unclear, and certainly not compelling.

My conclusion is that this paper does not support its conclusions with clear evidence and that it does not present enough that is new to merit publication.

---

## Referee Comment (RC2) · Anonymous Referee #2 · 10 Dec 2018

This paper is to prove the model for double substorm onsets which is thought to occur when the IMF turns from southward to northward, through the analysis of the data of the magnetic field and plasma obtained by THEMIS spacecraft, and those of aurora, geomagnetic field and geomagnetic pulsations on the ground, together with the solar wind data. However, it seems that the characteristics and variations in each data reported in this paper are considerably biased by the Russell model for double substorm onsets. Since the interpretation of the observational data are incomplete and misleading, it is a pity but I cannot recommend this paper for publication. I hope that the authors revise the paper taking the following comments into account, and submit it again.

P4, L6 - P5, L8: I do not understand the reason why the detailed explanations on the

event described in other paper are given here. You should explain the events during disturbed period in more detail here.

P4, L20-22: I'm not sure whether the timing for E2 is correct. Besides there are similar variations in the observed data other than E1 and E2.

P4, L23-25: You do not show any data for AL index for this example in the text.

P4, L32-33: Similarly to the above comment, any data for AL index for this example is not given.

P5, L10-22: The C2 onset does not look like a substorm. I don't understand the reason why you draw a line here. The bay-like variation is also understood as a variation simply associated with a recovery phase.

P5, L23: I cannot see the two-stage dipolarization as described in Fig. 3a.

P5, L24: The auroral expansion starting at the timing of C1 continues to C2, and it seems to disappear immediately after C2.

P5, L28: In any of the spacecraft data, there are no clear indications of dipolarizations for C1, C2.

P6, L4-6: Flow burst does not seem to occur in association with the # 1 onset.

P6, L11: What is the relationship between pseudosubstorms and IMF Bz?

P6, L16-21: The quality of the panels should be improved. The auroral variations described here are difficult to be read from the figure.

P6, L21-24: I understand that the aurora for the second onset develops more in the west, but I'm not sure it occurs on the higher latitude than the first onset aurora.

P6, L24-32: The quality of Fig. 7 is better than Fig.6, but still it should be improved to increase the contrast of the image and make your point clearer. What I noticed by looking at the present version of the figure is that the auroral arc brightened due to the

first onset seems to move to higher latitudes, and the same arc brightened again at the second onset. According to the Russell model, I think that the first onset arc brightens in a given latitude and the second onset arc should newly appear in higher latitudes.

P7, L2-20: It is more important to consider the location of the westward electrojet (or the wedge current) based on the distributions of H and D component variations rather than to describe the detailed geomagnetic field variations at various locations.

P7, L11-12: I don't see any intensification of the magnetic bay variations at the C1 onset.

P7, L22-P8, L19: Throughout the time of interest, the value of |By (OMNI)| is comparable to |Bz (OMNI)|. Don't you need to consider its effect in the present analysis?

P7, L31-32: It is true that the IMF turned to northward, but shortly after it turned to southward. Can you apply the Russell model to such a case?

P8, L11: Bz (OMNI) has turned to northward well before the #2 onset.

P8 L21 - P9, L4: The characters of the geomagnetic pulsations described here are not necessarily clear. Also, it should be noted that, since the authors compare the data obtained by induction magnetometer, they must divide the output from the wavelet analysis by wave frequency to obtain the wave intensity.

P10, L22-24: I do not understand the reason why the authors show the result of the correlation analysis here.

Additionally, I have the following general questions:

* Isn't it possible to show the precise timings of the auroral breakup, flow burst, dipolarization, geomagnetic bays and geomagnetic pulsations, etc. relative to the polarity change of the IMF within the time resolution of the employed data.

* I don't think that the IMF changes from southward to northward always results in double substorm onsets. What is the other conditions that cause the double substorm

onsets?

* If you want to show the validity of the Russell model, it is more appropriate to obtain the evidence that shows the quenching the magnetic connection at the DNL in the magnetotail when the IMF turns to northward and the associated decrease in plasma density in the lobe. Furthermore, it is useful to show the occurrence of the reconnection of the plasma sheet magnetic fields at the NENL for the first onset, and the reconnection of the lobe magnetic field in the second onset through the analysis of the spacecraft data.

There are also some minor comments:

P3, L30: The dates shown here seem to be incorrect.

P4, L1-4: Both LETH and FSI are not shown in Fig. 1.

P4, L7-9: The locations of the THEMIS spacecraft should be given.

P5, L16: The time shown here seems incorrect.

P21, Fig. 8a: Isn't it "CHBG" rather than "CHNG"?

---

## Referee Comment (RC3) · Anonymous Referee #3 · 17 Dec 2018

This paper studied two-step development of substorm expansion or two successive substorm expansions by examining fast earthward flows and dipolarization in the magnetotail from multipoint THEMIS spacecraft, auroral breakups, geomagnetic field bay changes, and Pi2 and lower-frequency pulsations from high and low latitude ground stations, and IMF changes for the four substorm events. The authors concluded that the first onset occurs during southward IMF, while the second one is caused by IMF northward turning.

The substorm onset and development mechanism is an important issue, and the present results may potentially give a clue to understanding of a substorm external trigger, i.e., IMF northward turning. I, however, do not think that the manuscript is well written, and hence I cannot recommend to accept this manuscript for publication

in its present form. The manuscript needs to be substantially revised by describing the observational results and related discussion in more detail, as I point out below. First, the authors state that each of the first and second onsets of the double-onset events had all of the magnetotail and ground signatures. Some onsets, however, do not seem to have all signatures, or some signatures are not clearly shown and specifically described. The reason for the lack of the signatures should be discussed, along with more detailed description of the observations. Second, the authors did not consider the propagation time of the effect of the solar wind and IMF changes from the bow shock nose or the spacecraft to the near-Earth tail and ground through the tail reconnection site. Without this consideration, the conclusion would not be convincing.

Specific comments:

Page 3, line 20: For the first and second events (quiet time events shown in Figs. 2 and 3), it is difficult to understand the locations of the THEMIS spacecraft and the relative locations of their footprints and the ground stations only from this manuscript. Currently the paper by Cheng et al. (submitted, 2018a) does not seem open, so the readers need to refer other sources by themselves to know the locations. I suggest to add figures and/or more detailed description of the locations to this paper.

Page 4, lines 11-14, and the captions of Figs. 2 and 5: It is not clear what the vertical lines in Figs. 2-5 indicate. Do they indicate the first time when the Vx component of the ion flow exceeded 100 km/s or the time when the earthward flow started to grow fast? Some vertical lines appear to correspond to either of these timings. Some other vertical lines, however, do not correspond to any fast earthward flows (e.g., D in Fig. 2b, C1 in Fig. 4b, and PS and #1 in Fig. 5). Furthermore, some other vertical lines are drawn at a later time, although they correspond to fast earthward flows (e.g., A in Fig. 3b, and #2 in Fig. 5b). Please make this point clear and check the timings of the vertical lines.

Figs. 2-5 keograms: What is the meridian for the keograms? The center of the field of

view?

Page 4, lines 22-24: The AL index as well as the IMF is not shown later in this paper for the 18 March 2009 event. Also, the AL index should grow (become more negative), not decline, associated with the second onset (E2), similar to the first onset (E1). The H component at SNAP show this negative bay change at E2 (Fig. 2c). This is the case with the events shown in Fig. 9.

Page 4, lines 30-31, and page 5, line 2 (Fig. 2c): Where were the SNAP, FSMI, and LETH stations located, relative to the footprints of the THEMIS spacecraft?

Page 4, line 31 to page 5, line 1: It is not clear which auroral activity the authors link to the second onset E2. That is, it does not seem to me that further poleward expansion occurred at a higher latitude at E2, although the negative bay was observed at SNAP then.

Page 5, line 3: The negative bay at SNAP began before E2, not after E2.

Page 5, line 17: Again, it is not clear which auroral activity the authors link to the second onset C2. That is, it does not seem to me that further poleward expansion occurred at a higher latitude at C2, although the negative bay was observed at SNAP then.

Page 5, lines 22-25: This statement of two-step development does not seem to be clearly supported by the auroral observations, because of the lack of auroral breakup or poleward expansion at the second onset, as mentioned above.

Page 5, line 28: No clear dipolarization, i.e., Bz increase and |Bx| decrease, seems to be observed by any THEMIS spacecraft at the second onset C2.

Page 5, lines 29-33, Fig. 4c: The keograms are difficult to see the auroral activity associated with the onsets, particularly B and C2. Is this due to too high maximum of the color scale or the meridians for the keograms different from the auroral activity? Otherwise, it seems that no aurora was activated at B and further poleward expansion was not clearly seen at the second onset C2.

Also, the H component of the geomagnetic field does not show two-step development. That is, bay changes did not newly start at C2, but the H component just continued to increase or decrease around C2.

Fig. 5c. The keograms are difficult to see the auroral activity. The color scale should be adjusted to see the auroral activity more easily.

Page 6, line 9: The description of the geomagnetic field changes is too rough. In particular, the H bays at RANK and GILL for #2 should be mentioned, since the authors focus on two-step development at #1 and #2.

Section 2.2: The authors did not describe the results of the geomagnetic pulsations for all of the present four events in this section. Since the authors regard the pulsations as further evidence of the double onsets, the onset times and characteristics of the pulsations, for example, should be described.

Furthermore, for discussion about the causal links between the onset signatures, it would be helpful to describe the more detailed relative timings between the onset signatures in the magnetotail (fast earthward flow and dipolarization) and on the ground (auroral breakup, geomagnetic field changes, and pulsation), with consideration of the relative locations of the spacecraft footprints and the ground stations. For example, for the first event (Fig. 2), the E1 activity began nearly at the same time as the flow burst, while the E2 negative bay at SNAP began 1 min to 30 s before the flow burst. Such relative timings and their explanation may be helpful for the interpretation of the events.

Page 7, lines 8 and 11: Were the unclear pulsations at DOB and DON real? Are these due to the lower time resolution of these data? (What is the time resolution of these data?)

Section 2.5: The definition of the IMF clock angle should be described. In addition, is the clock angle calculated correctly? For example, at ∼0220 UT in Fig. 9a, the OMNI IMF was directed almost duskward, while the Geotail IMF was duskward and

southward. Hence the clock angle difference between them should be 45 deg, but that in Fig. 9a is ∼180 deg. Furthermore, it seems strange that even the sign of them differ.

When the authors examined the correspondence between the IMF change and the substorm onset, they compared the IMF at the bow shock nose or the spacecraft (Geotail and Cluster) with the onset timing and did not consider the propagation time of the effect of IMF change from the bow shock nose or the spacecraft to the tail reconnection site as well as the propagation time of the reconnection effect from the tail reconnection site to the near-Earth tail and the ground. Considering these propagation times, it should be confirmed whether the IMF changes really correspond to each of the first and second onsets.

Other minor corrections:

Page 3, line 21: three –> four

Page 3, line 30: on 18 March 2009 and 3 April 2009 –> on 16 February 2008 and 24 February 2010

Fig. 1b: Add the labels of the vertical axis of the two panels.

Fig. 8a, bottom: CHNG –> CHBG

Fig. 11 caption, line 3: The vertical line –> The vertical solid line